# Recurrent Video Restoration Transformer with Guided Deformable Attention

Jingyun Liang[1], Yuchen Fan[2], Xiaoyu Xiang[2], Rakesh Ranjan[2], Eddy Ilg[2]
Simon Green[2], Jiezhang Cao[1], Kai Zhang[1]*, Radu Timofte[1,3], Luc Van Gool[1]

[1]Computer Vision Lab, ETH Zurich, Switzerland  [2]Meta Inc.  [3]University of Wurzburg, Germany

## Abstract

Video restoration aims at restoring multiple high-quality frames from multiple low-quality frames. Existing video restoration methods generally fall into two extreme cases, *i.e.*, they either restore all frames in parallel or restore the video frame by frame in a recurrent way, which would result in different merits and drawbacks. Typically, the former has the advantage of temporal information fusion. However, it suffers from large model size and intensive memory consumption; the latter has a relatively small model size as it shares parameters across frames; however, it lacks long-range dependency modeling ability and parallelizability. In this paper, we attempt to integrate the advantages of the two cases by proposing a recurrent video restoration transformer, namely RVRT. RVRT processes local neighboring frames in parallel within a globally recurrent framework which can achieve a good trade-off between model size, effectiveness, and efficiency. Specifically, RVRT divides the video into multiple clips and uses the previously inferred clip feature to estimate the subsequent clip feature. Within each clip, different frame features are jointly updated with implicit feature aggregation. Across different clips, the guided deformable attention is designed for clip-to-clip alignment, which predicts multiple relevant locations from the whole inferred clip and aggregates their features by the attention mechanism. Extensive experiments on video super-resolution, deblurring, and denoising show that the proposed RVRT achieves state-of-the-art performance on benchmark datasets with balanced model size, testing memory and runtime. The codes are available at https://github.com/JingyunLiang/RVRT.

## 1 Introduction

Video restoration, such as video super-resolution, deblurring, and denoising, has become a hot topic in recent years. It aims to restore a clear and sharp high-quality video from a degraded (*e.g.*, downsampled, blurred, or noisy) low-quality video [80, 11, 4, 38]. It has wide applications in live streaming [97], video surveillance [49], old film restoration [78], and more.

Parallel methods and recurrent methods have been dominant strategies for solving various video restoration problems. Typically, those two kinds of methods have their respective merits and demerits. Parallel methods [2, 25, 80, 72, 36, 63, 99, 27, 35, 4, 38] support distributed deployment and achieve good performance by directly fusing information from multiple frames, but they often have a large model size and consume enormous memory for long-sequence videos. In the meanwhile, recurrent models [24, 59, 21, 23, 26, 28, 9, 11, 45, 55, 98, 62] reuse the same network block to save parameters and predict the new frame feature based on the previously refined frame feature, but the sequential processing strategy inevitably leads to information loss and noise amplification [14] for long-range dependency modelling and makes it hard to be parallelized.

---

*Corresponding Author

36th Conference on Neural Information Processing Systems (NeurIPS 2022).

Considering the advantages and disadvantages of parallel and recurrent methods, in this paper, we propose a recurrent video restoration transformer (RVRT) that takes the best of both worlds. On the one hand, RVRT introduces the recurrent design into transformer-based models to reduce model parameters and memory usage. On the other hand, it processes neighboring frames together as a clip to reduce video sequence length and alleviate information loss. To be specific, we first divide the video into fixed-length video clips. Then, starting from the first clip, we refine the subsequent clip feature based on the previously inferred clip feature and the old features of the current clip from shallower layers. Within each clip, different frame features are jointly extracted, implicitly aligned and effectively fused by the self-attention mechanism [77, 51, 39]. Across different clips, information is accumulated clip by clip with a larger hidden state than previous recurrent methods.

To implement the above RVRT model, one big challenge is how to align different video clips when using the previous clip for feature refinement. Most existing alignment techniques [58, 65, 59, 87, 9, 15, 72, 80, 11, 38] are designed for frame-to-frame alignment. One possible way to apply them to clip-to-clip alignment is by introducing an extra feature fusion stage after aligning all frame pairs. Instead, we propose an one-stage video-to-video alignment method named guided deformable attention (GDA). More specifically, for a reference location in the target clip, we first estimate the coordinates of multiple relevant locations from different frames in the supporting clip under the guidance of optical flow, and then aggregate features of all locations dynamically by the attention mechanism.

GDA has several advantages over previous alignment methods: 1) Compared with optical flow-based warping that only samples one point from one frame [59, 87, 9], GDA benefits from multiple relevant locations sampled from the video clip. 2) Unlike mutual attention [38], GDA utilizes features from arbitrary locations without suffering from the small receptive field in local attention or the huge computation burden in global attention. Besides, GDA allows direct attention on non-integer locations with bilinear interpolation. 3) In contrast to deformable convolution [15, 100, 72, 80, 11, 10] that uses a fixed weight in feature aggregation, GDA generates dynamic weights to aggregate features from different locations. It also supports arbitrary location numbers and allows for both frame-to-frame and video-to-video alignment without any modification.

Our contributions can be summarized as follows:

- We propose the recurrent video restoration transformer (RVRT) that extracts features of local neighboring frames from one clip in a joint and parallel way, and refines clip features by accumulating information from previous clips and previous layers. By reducing the video sequence length and transmitting information with a larger hidden state, RVRT alleviates information loss and noise amplification in recurrent networks, and also makes it possible to partially parallelize the model.
- We propose the guided deformable attention (GDA) for one-stage video clip-to-clip alignment. It dynamically aggregates information of relevant locations from the supporting clip.
- Extensive experiments on eight benchmark datasets show that the proposed model achieves state-of-the-art performance in three challenging video restoration tasks: video super-resolution, video deblurring, and video denoising, with balanced model size, memory usage and runtime.

## 2 Related Work

### 2.1 Video Restoration

**Parallel vs. recurrent methods.** Most existing video restoration methods can be classified as parallel or recurrent methods according to their parallelizability. Parallel methods estimate all frames simultaneously, as the refinement of one frame feature is not dependent on the update of other frame features. They can be further divided as sliding window-based methods [2, 25, 80, 70, 72, 79, 36, 63, 99, 99, 27, 71, 60, 35] and transformer-based methods [4, 38]. The former kind of methods typically restore merely the center frame from the neighboring frames and are often tested in a sliding window fashion rather than in parallel. These methods generally consist of four stages: feature extraction, feature alignment, feature fusion, and frame reconstruction. Particularly, in the feature alignment stage, they often align all frames towards the center frame, which leads to quadratic complexity with respect to video length and is hard to be extended for long-sequence videos. Instead, the latter kind of method reconstructs all frames at a time based on the transformer architectures. They jointly extract, align, and fuse features for all frames, achieving significant

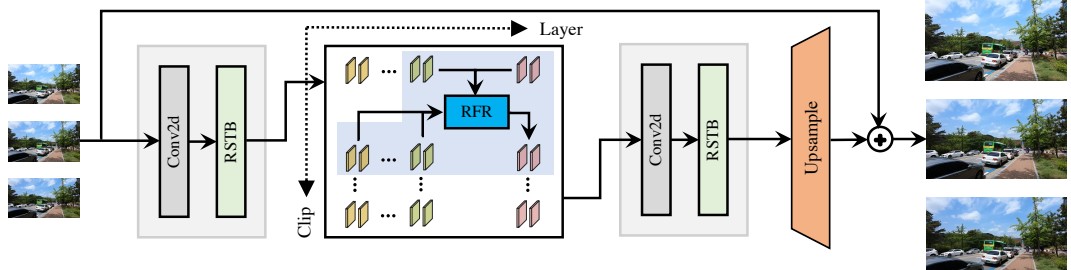

Figure 1: The architecture of recurrent video restoration transformer (RVRT). From left to right, it consists of shallow feature extraction, recurrent feature refinement and HQ frame reconstruction. In recurrent feature refinement (RFR, see more details in Fig. 2), we divide the video into $N$-frame clips ($N = 2$ in this figure) and process frames in one clip in parallel within a globally recurrent framework in time. Multiple refinement layers are stacked for better performance.

performance improvements against previous methods. However, current transformer-based methods are laid up with a huge model size and large memory consumption. Different from above parallel methods, recurrent methods [24, 59, 21, 23, 85, 26, 28, 9, 11, 45, 55, 98, 62, 44, 5] propagate latent features from one frame to the next frame sequentially, where information of previous frames is accumulated for the restoration of later frames. Basically, they are composed of three stages: feature extraction, feature propagation and frame reconstruction. Due to the recurrent nature of feature propagation, recurrent methods suffer from information loss and the inapplicability of distributed deployment.

**Alignment in video restoration.** Unlike image restoration that mainly focuses on feature extraction [16, 94–96, 42, 40, 41, 67, 93, 92], how to align multiple highly-related but misaligned frames is another key problem in video restoration. Traditionally, many methods [43, 30, 2, 48, 68, 3, 87, 9] first estimate the optical flow between neighbouring frames [19, 58, 65] and then conduct image warping for alignment. Other techniques, such as deformable convolution [15, 100, 72, 80, 11, 4], dynamic filter [29] and mutual attention [38], have also been exploited for implicit feature alignment.

## 2.2 Vision Transformer

Transformer [77] is the de-facto standard architecture in natural language processing. Recently, it has been used in dealing with vision problems by viewing pixels or image patches as tokens [8, 18], achieving remarkable performance gains in various computer vision tasks, including image classification [18, 37, 51, 74], object detection [76, 50, 84], semantic segmentation [83, 17, 66], *etc*. It also achieves promising results in restoration tasks [13, 39, 81, 44, 4, 38, 20, 22, 7, 90, 47, 73, 6]. In particular, for video restoration, Cao *et al*. [4] propose the first transformer model for video SR, while Liang *et al*. [38] propose an unified framework for video SR, deblurring and denoising.

We note that some transformer-based works [101, 84] have tried to combine the concept of deformation [15, 100] with the attention mechanism [77]. Zhu *et al*. [101] directly predicts the attention weight from the query feature without considering its feature interaction with supporting locations. Xia *et al*. [84] place the supporting points uniformly on the image to make use of global information. Both above two methods are proposed for recognition tasks such as object detection, which is fundamentally different from video alignment in video restoration. Lin *et al*. [44] use pixel-level or patch-level attention to aggregate information from neighbouring frames under the guidance of optical flow, but it only samples one supporting pixel or patch from one frame, restricting the model from attending to multiple distant locations.

## 3 Methodology

### 3.1 Overall Architecture

Given a low-quality video sequence $I^{LQ} \in \mathbb{R}^{T \times H \times W \times C}$, where $T$, $H$, $W$ and $C$ are the video length, height, width and channel, respectively, the goal of video restoration is to reconstruct the high-quality video $I^{HQ} \in \mathbb{R}^{T \times sH \times sW \times C}$, where $s$ is the scale factor. To reach this goal, we propose a recurrent

video restoration transformer, as illustrated in Fig. 1. The model consists of three parts: shallow feature extraction, recurrent feature refinement and HQ frame reconstruction. More specifically, in shallow feature extraction, we first use a convolution layer to extract features from the LQ video. For deblurring and denoising (*i.e.*, $s = 1$), we additionally add two strided convolution layers to downsample the feature and reduce computation burden in the next layers. After that, several Residual Swin Transformer Blocks (RSTBs) [39] are used to extract the shallow feature. Then, we use recurrent feature refinement modules for temporal correspondence modeling and guided deformable attention for video alignment, which are detailed in Sec. 3.2 and Sec. 3.3, respectively. Lastly, we add several RSTBs to generate the final feature and reconstruct the HQ video $I^{RHQ}$ by pixel shuffle layer [61]. For training, the Charbonnier loss [12] $\mathcal{L} = \sqrt{\|I^{RHQ} - I^{HQ}\|^2 + \epsilon^2}$ ($\epsilon = 10^{-3}$) is used for all tasks.

## 3.2 Recurrent Feature Refinement

We stack $L$ recurrent feature refinement modules to refine the video feature by exploiting the temporal correspondence between different frames. To make a trade-off between recurrent and transformer-based methods, we process $N$ frames locally in parallel on the basis of a globally recurrent framework.

Formally, given the video feature $F^i \in \mathbb{R}^{T \times H \times W \times C}$ from the $i$-th layer, we first reshape it as a 5-dimensional tensor $F^i \in \mathbb{R}^{\frac{T}{N} \times N \times H \times W \times C}$ by dividing it into $\frac{T}{N}$ video clip features: $F_1^i, F_2^i, ..., F_{\frac{T}{N}}^i \in \mathbb{R}^{N \times H \times W \times C}$. Each clip feature $F_t^i$ ($1 \leq t \leq \frac{T}{N}$) has $N$ neighbouring frame features: $F_{t,1}^i, F_{t,2}^i, ..., F_{t,N}^i \in \mathbb{R}^{H \times W \times C}$. To utilize information from neighbouring clips, we align the $(t-1)$-th clip feature $F_{t-1}^i$ towards the $t$-th clip based on the optical flow $O_{t-1 \rightarrow t}^i$, clip feature $F_{t-1}^{i-1}$ and clip feature $F_t^{i-1}$. This is formulated as follows:

$$\widehat{F}_{t-1}^i = GDA(F_{t-1}^i; O_{t-1 \rightarrow t}^i, F_{t-1}^{i-1}, F_t^{i-1}), \quad (1)$$

where *GDA* is the guided deformable attention and $\widehat{F}_{t-1}^i$ is the aligned clip feature. The details of GDA will be described in Sec. 3.3.

Similar to recurrent neural networks [59, 9, 11], as shown in Fig. 2, we update the clip feature of each time step as follows:

$$F_t^i = RFR(F_t^0, F_t^1, ..., F_t^{i-1}, \widehat{F}_{t-1}^i), \quad (2)$$

where $F_t^0$ is the output of the shallow feature extraction module and $F_t^1, F_t^2, ..., F_t^{i-1}$ are from previous recurrent feature refinement modules. $RFR(\cdot)$ is the recurrent feature refinement module that consists of a convolution layer for feature fusion and several modified residual Swin Transformer blocks (*MRSTBs*) for feature refinement. In *MRSTB*, we upgrade the original 2D $h \times w$ attention window to the 3D $N \times h \times w$ attention window, so that every frame in the clip can attend to itself and other frames simultaneously, allowing implicit feature aggregation. In addition, in order to accumulate information forward and backward in time, we reverse the video sequence for all even recurrent feature refinement modules [24, 11].

Figure 2: The illustrations of recurrent feature refinement (RFR). The $(t-1)$-th clip feature $F_{t-1}^i$ from the $i$-th layer is aligned towards the $t$-th clip as $\widehat{F}_{t-1}^i$ by guided deformable attention (GDA, see more details in Fig. 3). $F_t^0, F_t^1, ..., F_t^{i-1}$ and $\widehat{F}_{t-1}^i$ are then refined as $F_t^i$ by several modified residual swin transformer blocks (MRSTBs), in which different frames are jointly processed in a parallel way.

The above recurrent feature refinement module is the key component of the proposed RVRT model. Globally, features of different video clips are propagated in a recurrent way. Locally, features of different frames are updated jointly in parallel. For an arbitrary single frame, it can make full use of global information accumulated in time and local information extracted together by the self-attention mechanism. As we can see, RVRT is a generalization of both recurrent and transformer models. It becomes a recurrent model when $N = 1$ or a transformer model when $N = T$. This is fundamentally different from previous methods that adopt transformer blocks to replace CNN blocks within a recurrent architecture [78, 44]. It is also different from existing attempts in natural language processing [82, 34].

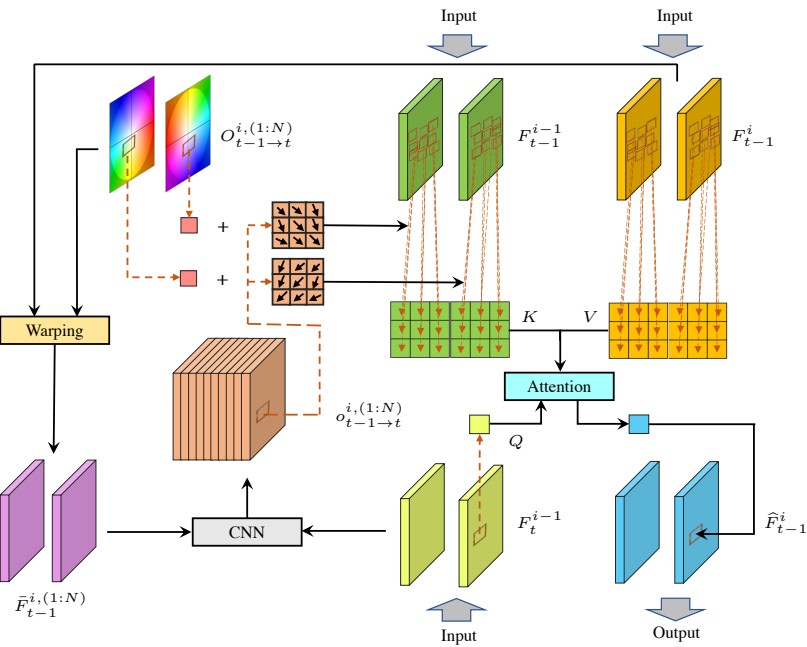

Figure 3: The illustrations of guided deformable attention (GDA). We estimate offsets of multiple relevant locations from different frames based on the warped clip, and then aggregate features of different locations dynamically by the attention mechanism. $F_{t-1}^i$ is the $(t-1)$-th clip feature from the $i$-th layer, while $\bar{F}_{t-1}^i$ and $\widehat{F}_{t-1}^i$ are the pre-aligned and aligned features of $F_{t-1}^i$. $O_{t-1\rightarrow t}^{i,(1:N)}$ and $o_{t-1\rightarrow t}^{i,(1:N)}$ denote optical flows and offsets, respectively.

## 3.3   Guided Deformable Attention for Video Alignment

Different from previous frameworks, the proposed RVRT needs to align neighboring related but misaligned video clips, as indicated in Eq. (1). In this subsection, we propose the guided deformation attention (GDA) for video clip-to-clip alignment.

Given the $(t-1)$-th clip feature $F_{t-1}^i$ from the $i$-th layer, our goal is to align $F_{t-1}^i$ towards the $t$-th clip as a list of features $\widehat{F}_{t-1}^{i,(1:N)} = \widehat{F}_{t-1}^{i,(1)}, \widehat{F}_{t-1}^{i,(2)}, ..., \widehat{F}_{t-1}^{i,(N)}$, where $\widehat{F}_{t-1}^{i,(n)}(1 \leq n \leq N)$ denotes the aligned clip feature towards the $n$-th frame feature $F_{t,n}^i$ of the $t$-th clip, and $\widehat{F}_{t-1,n'}^{i,(n)}(1 \leq n' \leq N)$ is the aligned frame feature from the $n'$-th frame in the $(t-1)$-th clip to the $n$-th frame in the $t$-th clip. Inspired by optical flow estimation designs [19, 56, 65, 11, 44], we first pre-align $F_{t-1,n'}^{i,(n)}$ with the optical flow $O_{t-1\rightarrow t,n'}^{i,(n)}$ as $\bar{F}_{t-1,n'}^{i,(n)} = \mathcal{W}(F_{t-1,n'}^{i,(n)}, O_{t-1\rightarrow t,n'}^{i,(n)})$, where $\mathcal{W}$ denotes the warping operation. For convenience, we summarize the pre-alignments of all "$n'$-to-$n$" $(1 \leq n', n \leq N)$ frame pairs between the $(t-1)$-th and $t$ video clips as follows:

$$\bar{F}_{t-1}^{i,(1:N)} = \mathcal{W}(F_{t-1}^i, O_{t-1\rightarrow t}^{i,(1:N)}), \tag{3}$$

After that, we predict the optical flow offsets $o_{t-1\rightarrow t}^{i,(1:N)}$ from the concatenation of $F_t^{i-1}$, $\bar{F}_{t-1}^{i,(1:N)}$ and $O_{t-1\rightarrow t}^{i,(1:N)}$ along the channel dimension. A small convolutional neural network (CNN) with several convolutional layers and ReLU layers is used for prediction. This is formulated as

$$o_{t-1\rightarrow t}^{i,(1:N)} = CNN(Concat(F_t^{i-1}, \bar{F}_{t-1}^{i,(1:N)}, O_{t-1\rightarrow t}^{i,(1:N)})), \tag{4}$$

where the current misalignment between the $t$-th clip feature and the warped $(t-1)$-th clip features can reflect the offset required for further alignment. In practice, we initialize $O_{t-1\rightarrow t}^{1,(1:N)}$ as the optical flows estimated from the LQ input video via SpyNet [58], and predict $M$ offsets for each frame ($NM$ offsets in total). The optical flows are updated layer by layer as follows:

$$O_{t-1\rightarrow t,n'}^{i+1,(n)} = O_{t-1\rightarrow t,n'}^{i,(n)} + \frac{1}{M}\sum_{m=1}^{M}\{o_{t-1\rightarrow t,n'}^{i,(n)}\}_m, \tag{5}$$

where $\{o^{i,(n)}_{t-1\to t,n'}\}_m$ denotes the $m$-th offset in $M$ predictions from the $n'$-th frame to the $n$-th frame.

Then, for the $n$-th frame of the $t$-th clip, we sample its relevant features from the $(t-1)$-th clip feature $F^i_{t-1}$ according the predicted locations, which are indicated by the sum of optical flow and offsets, *i.e.*, $O^{i,(n)}_{t-1\to t} + o^{i,(n)}_{t-1\to t}$, according to the chain relationship $F^i_{t-1} \xrightarrow{O^{i,(n)}_{t-1\to t}} \bar{F}^{i,(n)}_{t-1} \xrightarrow{o^{i,(n)}_{t-1\to t}} \widehat{F}^{i,(n)}_{t-1}$ [11, 58]. For simplicity, we define the queries $Q$, keys $K$ and values $V$ as follows:

$$Q = F^{i-1}_{t,n} P_Q, \tag{6}$$

$$K = Sampling(F^{i-1}_{t-1} P_K, O^{i,(n)}_{t-1\to t} + o^{i,(n)}_{t-1\to t}), \tag{7}$$

$$V = Sampling(F^i_{t-1} P_V, O^{i,(n)}_{t-1\to t} + o^{i,(n)}_{t-1\to t}), \tag{8}$$

where $Q \in \mathbb{R}^{1\times C}$ is the projected feature from the $n$-th frame of $t$-th clip. $K \in \mathbb{R}^{NM\times C}$ and $V \in \mathbb{R}^{NM\times C}$ are the projected features that are bilinearly sampled from $NM$ locations of $F^{i-1}_{t-1}$ and $F^i_{t-1}$, respectively. $P_Q \in \mathbb{R}^{C\times C}$, $P_K \in \mathbb{R}^{C\times C}$ and $P_V \in \mathbb{R}^{C\times C}$ are the projection matrices. Note that we first project the feature and then do sampling to reduce redundant computation.

Next, similar to the attention mechanism [77], we calculate the attention weights based on the $Q$ and $K$ from the $(i-1)$-th layer and then compute the aligned feature $\widehat{F}^{i,(n)}_{t-1}$ as a weighted sum of $V$ from the same $i$-th layer as follows:

$$\widehat{F}^{i,(n)}_{t-1} = SoftMax(QK^T/\sqrt{C}))V, \tag{9}$$

where *SoftMax* is the softmax operation along the row direction and $\sqrt{C}$ is a scaling factor.

Lastly, since Eq. (9) only aggregates information spatially, we add a multi-layer perception (*MLP*) with two fully-connected layers and a *GELU* activation function between them to enable channel interaction as follows:

$$\widehat{F}^i_{t-1} = \widehat{F}^i_{t-1} + MLP(\widehat{F}^i_{t-1}), \tag{10}$$

where a residual connection is used to stabilize training. The hidden and output channel numbers of the *MLP* are $RC$ ($R$ is the ratio) and $C$, respectively.

**Multi-group multi-head guided deformable attention.** We can divide the channel into several deformable groups and perform the deformable sampling for different groups in parallel. Besides, in the attention mechanism, we can further divide one deformable group into several attention heads and perform the attention operation separately for different heads. All groups and heads are concatenated together before channel interaction.

**Connection to deformable convolution.** Deformable convolution [15, 100] uses a learned weight for feature aggregation, which can be seen as a special case of GDA, *i.e.*, using different projection matrix $P_V$ for different locations and then directly averaging the resulting features. Its parameter number and computation complexity are $MC^2$ and $\mathcal{O}(MC^2)$, respectively. In contrast, GDA uses the same projection matrix for all locations but generates dynamic weights to aggregate them. Its parameter number and computation complexity are $(3+2R)C^2$ and $\mathcal{O}((3C+2RC+M)C)$, which are similar to deformable convolution when choosing proper $M$ and $R$.

## 4 Experiments

### 4.1 Experimental Setup

For shallow feature extraction and HQ frame reconstruction, we use 1 RSTB that has 2 swin transformer layers. For recurrent feature refinement, we use 4 refinement modules with a clip size of 2, each of which has 2 MRSTBs with 2 modified swin transformer layers. For both RSTB and MRSTB, spatial attention window size and head number are $8 \times 8$ and 6, respectively. We use 144 channels for video SR and 192 channels for deblurring and denoising. In GDA, we use 12 deformable groups and 12 deformable heads with 9 candidate locations. We empirically project the query to a higher-dimensional space (*e.g.*, $2C$) because we found it can improve the performance slightly and the parameter number of GDA is not a bottleneck. In training, we randomly crop $256 \times 256$ HQ patches and use different video lengths for different datasets: 30 frames for REDS [53], 14 frames for Vimeo-90K [87], and 16 frames for DVD [63], GoPro [54] as well as DAVIS [31]. Adam

optimizer [33] with default setting is used to train the model for 600,000 iterations when the batch size is 8. The learning rate is initialized as $4 \times 10^{-4}$ and deceased with the Cosine Annealing scheme [52]. To stabilize training, we initialize SpyNet [58, 56] with pretrained weights, fix it for the first 30,000 iterations and reduce its learning rate by 75%.

## 4.2 Ablation Study

To explore the effectiveness of different components, we conduct ablation studies on REDS [53] for video SR. For efficiency, we reduce the MRSTB blocks by half and use 12 frames in training.

**The impact of clip length.** In RVRT, we divide the video into $N$-frame clips. As shown in Table 1, the performance rises when clip length is increased from 1 to 2. However, the performance saturates when $N = 3$, possibly due to large within-clip motions and inaccurate optical flow derivation. When we directly estimate all optical flows (marked by $^*$), the PSNR hits 32.21dB. Besides, to compare the temporal modelling ability, we hack the input LQ video (Clip 000 from REDS, 100 frames in total) by manually setting all pixels of the 50-th frame as zeros. As indicated in Fig. 4, on the one hand, $N = 2$ has a smaller performance drop and all its frames still have higher PSNR than $N = 1$ (equals to a recurrent model) after the attack, showing that RVRT can mitigate the noise amplification from the hacked frame to the rest frames. One the other hand, the hacked frame of $N = 2$ has an impact on more neighbouring frames than $N = 1$, which means that RVRT can alleviate information loss and utilize more frames than $N = 1$ for restoration.

**The impact of video alignment.** The alignment of video clips plays a key role in our framework. We compare the proposed clip-to-clip guided deformable attention (GDA) with existing frame-to-frame alignment techniques by performing them frame by frame, followed by concatenation and channel reduction. As we can see from Table 2, GDA outperforms all existing methods when it is used for frame-to-frame alignment (denoted as GDA$^*$), and leads a further improvement when we aggregate features directly from the whole clip.

**The impact of different components in GDA.** We further conduct an ablation study on GDA in Table 3. As we can see, the optical flow guidance is critical for the model, leading to a PSNR gain of 1.11dB. The update of optical flow in different layers can further improve the result. The channel interaction in MLP also plays an important role, since the attention mechanism only aggregates information spatially.

**The impact of deformable group and attention head.** We also conduct experiments on different group and head numbers in GDA. As shown in Table 4, when the deformable group rises, the PSNR first rises and then keeps almost unchanged. Besides, double attention heads lead to slightly better results at the expense of higher computation, but using too many heads has an adverse impact as the head dimension may be too small.

Table 1: Ablation study on clip length.     Table 2: Ablation study on different video alignment techniques.

| Clip | 1 | 2 | 3 | 3$^*$ |
|------|------|------|------|------|
| PSNR | 31.98 | 32.10 | 32.07 | 32.21 |

| Alignment | Warping [87] | TMSA [38] | DCN [72] | GDA$^*$ | GDA |
|-----------|------|------|------|------|------|
| PSNR | 28.88 | 30.45 | 31.93 | 32.00 | 32.10 |

Figure 4: Per-frame PSNR drop when pixels of the 50-th frame is hacked to be all zeros. $N$ is clip length.

| Optical Flow Guidance | | ✓ | ✓ | ✓ |
|-----------------------|------|------|------|------|
| Optical Flow Update | | | ✓ | ✓ |
| MLP | ✓ | ✓ | | ✓ |
| PSNR | 30.99 | 32.03 | 31.83 | 32.10 |

Table 3: Ablation study on different GDA components.

| Deformable Group | 1 | 6 | 12 | 12 | 12 | 24 |
|------------------|------|------|------|------|------|------|
| Attention Head | 1 | 6 | 12 | 24 | 36 | 24 |
| PSNR | 31.63 | 32.03 | 32.10 | 32.13 | 32.03 | 32.11 |

Table 4: Ablation study on deformable groups and attention heads.

## 4.3 Video Super-Resolution

For video SR, we consider two settings: bicubic (BI) and blur-downsampling (BD) degradation. For BI degradation, we train the model on two different datasets: REDS [53] and Vimeo-90K [87],

Table 5: Quantitative comparison (average PSNR/SSIM) with state-of-the-art methods for **video super-resolution** ($\times 4$) on **REDS4** [53], **Vimeo-90K-T** [87], **Vid4** [46] and **UDM10** [89].

| Method | BI degradation | | | BD degradation | | |
|---|---|---|---|---|---|---|
| | REDS4 [53] (RGB channel) | Vimeo-90K-T [87] (Y channel) | Vid4 [46] (Y channel) | UDM10 [89] (Y channel) | Vimeo-90K-T [87] (Y channel) | Vid4 [46] (Y channel) |
| Bicubic | 26.14/0.7292 | 31.32/0.8684 | 23.78/0.6347 | 28.47/0.8253 | 31.30/0.8687 | 21.80/0.5246 |
| SwinIR [39] | 29.05/0.8269 | 35.67/0.9287 | 25.68/0.7491 | 35.42/0.9380 | 34.12/0.9167 | 25.25/0.7262 |
| SwinIR-ft [39] | 29.24/0.8319 | 35.89/0.9301 | 25.69/0.7488 | 36.76/0.9467 | 35.70/0.9293 | 25.62/0.7498 |
| TOFlow [87] | 27.98/0.7990 | 33.08/0.9054 | 25.89/0.7651 | 36.26/0.9438 | 34.62/0.9212 | 25.85/0.7659 |
| FRVSR [59] | - | - | - | 37.09/0.9522 | 35.64/0.9319 | 26.69/0.8103 |
| DUF [29] | 28.63/0.8251 | - | 27.33/0.8319 | 38.48/0.9605 | 36.87/0.9447 | 27.38/0.8329 |
| PFNL [89] | 29.63/0.8502 | 36.14/0.9363 | 26.73/0.8029 | 38.74/0.9627 | - | 27.16/0.8355 |
| RBPN [23] | 30.09/0.8590 | 37.07/0.9435 | 27.12/0.8180 | 38.66/0.9596 | 37.20/0.9458 | 27.17/0.8205 |
| MuCAN [36] | 30.88/0.8750 | 37.32/0.9465 | - | - | - | - |
| RLSP [21] | - | - | - | 38.48/0.9606 | 36.49/0.9403 | 27.48/0.8388 |
| TGA [27] | - | - | - | 38.74/0.9627 | 37.59/0.9516 | 27.63/0.8423 |
| RSDN [26] | - | - | - | 39.35/0.9653 | 37.23/0.9471 | 27.92/0.8505 |
| RRN [28] | - | - | - | 38.96/0.9644 | - | 27.69/0.8488 |
| FDAN [45] | - | - | - | 39.91/0.9686 | 37.75/0.9522 | 27.88/0.8508 |
| EDVR [80] | 31.09/0.8800 | 37.61/0.9489 | 27.35/0.8264 | 39.89/0.9686 | 37.81/0.9523 | 27.85/0.8503 |
| GOVSR [88] | - | - | - | 40.14/0.9713 | 37.63/0.9503 | 28.41/0.8724 |
| BasicVSR [9] | 31.42/0.8909 | 37.18/0.9450 | 27.24/0.8251 | 39.96/0.9694 | 37.53/0.9498 | 27.96/0.8553 |
| IconVSR [9] | 31.67/0.8948 | 37.47/0.9476 | 27.39/0.8279 | 40.03/0.9694 | 37.84/0.9524 | 28.04/0.8570 |
| VRT [38] | 32.19/0.9006 | 38.20/0.9530 | 27.93/0.8425 | 41.05/0.9737 | 38.72/0.9584 | 29.42/0.8795 |
| BasicVSR++ [11] | 32.39/0.9069 | 37.79/0.9500 | 27.79/0.8400 | 40.72/0.9722 | 38.21/0.9550 | 29.04/0.8753 |
| **RVRT** (ours) | 32.75/0.9113 | 38.15/0.9527 | 27.99/0.8462 | 40.90/0.9729 | 38.59/0.9576 | 29.54/0.8810 |

Table 6: Comparison of model size, testing memory and runtime for an LQ input of $320 \times 180$.

| Method | #Param (M) | Memory (M) | Runtime (ms) | PSNR (dB) |
|---|---|---|---|---|
| BasicVSR++ [11] | 7.3 | 223 | 77 | 32.39 |
| BasicVSR++ [11]+RSTB [39] | 9.3 | 1021 | 201 | 32.61 |
| EDVR [80] | 20.6 | 3535 | 378 | 31.09 |
| VSRT [4] | 32.6 | 27487 | 328 | 31.19 |
| VRT [38] | 35.6 | 2149 | 243 | 32.19 |
| **RVRT** (ours) | 10.8 | 1056 | 183 | 32.75 |

and then test the model on their corresponding testsets: REDS4 and Vimeo-90K-T. We additionally test Vid4 [46] along with Vimeo-90K. For BD degradation, we train it on Vimeo-90K and test it on Vimeo-90K-T, Vid4, and UDM10 [89]. The comparisons with existing methods are shown in Table 5. As we can see, RVRT achieves the best performance on REDS4 and Vid4 for both degradations. Compared with the representative recurrent model BasicVSR++ [11], RVRT improves the PSNR by significant margins of **0.2~0.5dB**. Compared with the recent transformer-based model VRT [38], RVRT outperforms VRT on REDS4 and Vid4 by **up to 0.36dB**. The visual comparisons of different methods are shown in Fig. 5. It is clear that RVRT generates sharp and clear HQ frames, while other methods fail to restore fine textures and details.

We compare the model size, testing memory consumption and runtime of different models in Table 6. Compared with representative parallel methods EDVR [80], VSRT [4] and VST [38], RVRT achieves significant performance gains with less than **at least 50%** of model parameters and testing memory usage. It also reduces the runtime by **at least 25%**. Compared the recurrent model BasicVSR++ [11], RVRT brings a PSNR improvement of 0.26dB. As for the inferiority of testing memory and runtime, we argue that it is mainly because the CNN layers are highly optimized on existing deep learning frameworks. To prove it, we use the transformer-based RSTB blocks in RVRT to replace the CNN blocks in BasicVSR++, in which case it has similar memory usage and more runtime than our model.

In addition, to better understand how guided deformable attention works, we visualize the predicted offsets on the LQ frames and show the attention weight in Fig. 6. As we can see, multiple offsets are predicted to select multiple sampled locations in the neighbourhood of the corresponding pixel. According to the feature similarity between the query feature and the sampled features, features of different locations are aggregated by calculating a dynamic attention weight.

## 4.4 Video Deblurring

For video deblurring, the model is trained and tested on two different datasets, DVD [63] and GoPro [54], with their official training/testing splits. As shown in Table 7 and 8, RVRT shows its

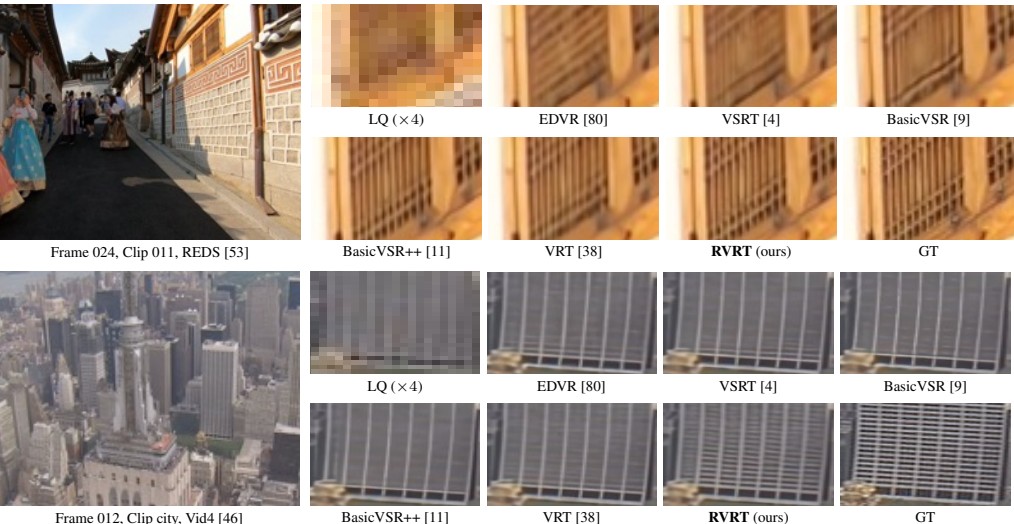

Figure 5: Visual comparison of **video super-resolution** ($\times 4$) methods on REDS [53] and Vid4 [46].

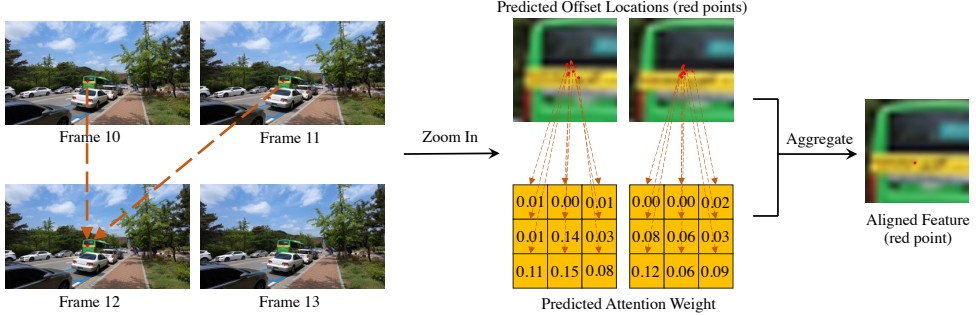

Figure 6: The visualization of predicted offsets and attention weight predicted in guided deformable attention. Although guided deformable attention is conducted on features, we plot illustrations on LQ input frames for better understanding. Best viewed by zooming.

superiority over most methods with huge improvements of **1.40∼2.27dB** on two datasets. Even though the performance gain over VRT is relatively small, RVRT has a smaller model size and much less runtime. In detail, the model size and runtime of RVRT are 13.6M and 0.3s, while VRT has 18.3M parameters and the runtime of 2.2s on a $1280 \times 720$ LQ input. The visual comparison is provided in the supplementary material due to the space limit.

## 4.5 Video Denoising

For video denoising, we train the model on the training set of DAVIS [31] and test it on its corresponding testset and Set8 [70]. For fairness of comparison, following [70, 71], we train a non-blind additive white Gaussian denoising model for noise level $\sigma \sim \mathcal{U}(0, 50)$. Similar to the case of video deblurring, there is a huge gap (**0.60∼2.37dB**) between RVRT and most methods. Compared with VRT, RVRT has slightly better performance on large noise levels, with a smaller model size (12.8M *v.s.*18.4M) and less runtime (0.2s *v.s.*1.5s) on a $1280 \times 720$ LQ input. The visual comparison is provided in the supplementary material due to the space limit.

## 5 Conclusion

In this paper, we proposed a recurrent video restoration transformer with guided deformable attention. It is a globally recurrent model with locally parallel designs, which benefits from the advantages of both parallel methods and recurrent methods. We also propose the guided deformable attention

Table 7: Quantitative comparison (average RGB channel PSNR/SSIM) with state-of-the-art methods for **video deblurring** on **DVD** [63].

| Method | DBN [63] | STFAN [99] | STTN [32] | SFE [86] | EDVR [80] | TSP [57] |
|---|---|---|---|---|---|---|
| PSNR | 30.01 | 31.24 | 31.61 | 31.71 | 31.82 | 32.13 |
| SSIM | 0.8877 | 0.9340 | 0.9160 | 0.9160 | 0.9160 | 0.9268 |
| Method | PVDNet [62] | GSTA [64] | ARVo [35] | FGST [44] | VRT [38] | **RVRT** (ours) |
| PSNR | 32.31 | 32.53 | 32.80 | 33.36 | 34.24 | 34.30 |
| SSIM | 0.9260 | 0.9468 | 0.9352 | 0.9500 | 0.9651 | 0.9655 |

Table 8: Quantitative comparison (average RGB channel PSNR/SSIM) with state-of-the-art methods for **video deblurring** on **GoPro** [54].

| Method | SRN [69] | MPRNet [91] | MAXIM [73] | IFI-RNN [55] | ESTRNN [98] | EDVR [80] |
|---|---|---|---|---|---|---|
| PSNR | 30.26 | 32.66 | 32.86 | 31.05 | 31.07 | 31.54 |
| SSIM | 0.9342 | 0.9590 | 0.9610 | 0.9110 | 0.9023 | 0.9260 |
| Method | TSP [57] | PVDNet [62] | GSTA [64] | FGST [44] | VRT [38] | **RVRT** (ours) |
| PSNR | 31.67 | 31.98 | 32.10 | 32.90 | 34.81 | 34.92 |
| SSIM | 0.9279 | 0.9280 | 0.9600 | 0.9610 | 0.9724 | 0.9738 |

Table 9: Quantitative comparison (average RGB channel PSNR) with state-of-the-art methods for **video denoising** on **DAVIS** [31] and **Set8** [70].

| Dataset | $\sigma$ | VLNB [1] | DVDNet [70] | FastDVDNet [71] | PaCNet [75] | VRT [38] | **RVRT** (ours) |
|---|---|---|---|---|---|---|---|
| DAVIS | 10 | 38.85 | 38.13 | 38.71 | 39.97 | 40.82 | 40.57 |
| | 20 | 35.68 | 35.70 | 35.77 | 36.82 | 38.15 | 38.05 |
| | 30 | 33.73 | 34.08 | 34.04 | 34.79 | 36.52 | 36.57 |
| | 40 | 32.32 | 32.86 | 32.82 | 33.34 | 35.32 | 35.47 |
| | 50 | 31.13 | 31.85 | 31.86 | 32.20 | 34.36 | 34.57 |
| Set8 | 10 | 37.26 | 36.08 | 36.44 | 37.06 | 37.88 | 37.53 |
| | 20 | 33.72 | 33.49 | 33.43 | 33.94 | 35.02 | 34.83 |
| | 30 | 31.74 | 31.79 | 31.68 | 32.05 | 33.35 | 33.30 |
| | 40 | 30.39 | 30.55 | 30.46 | 30.70 | 32.15 | 32.21 |
| | 50 | 29.24 | 29.56 | 29.53 | 29.66 | 31.22 | 31.33 |

module for our special case of video clip-to-clip alignment. Under the guidance of optical flow, it aggregates information from multiple neighboring locations adaptively with the attention mechanism. Extensive experiments on video super-resolution, video deblurring, and video denoising demonstrated the effectiveness of the proposed method.

## 6   Limitations and Societal Impacts

Although RVRT achieves state-of-the-art performance in video restoration, it still has some limitations. For example, the complexity of pre-alignment by optical flow increases quadratically with respect to the clip length. One possible solution is to develop a video-to-video optical flow estimation model that directly predicts all optical flows. As for societal impacts, similar to other restoration methods, RVRT may bring privacy concerns after restoring blurry videos and lead to misjudgments if used for medical diagnosis.

## Acknowledgments and Disclosure of Funding

This work was partially supported by the ETH Zurich Fund (OK), a Huawei Technologies Oy (Finland) project, the China Scholarship Council and an Amazon AWS grant. Special thanks goes to Yijue Chen.

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
