# Supplementary Material for Recurrent Video Restoration Transformer with Guided Deformable Attention

In this supplementary material, we first give more details on training and testing datasets, as well as evaluation metrics. Then, we provide more visual comparisons of different methods.

## 1 More Details on Datasets and Evaluation Metrics

**Training and testing datasets.** For video super-resolution, we train the model on two different training datasets for scale factor 4. First, we generate low-resolution images by the MATLAB `imresize` function (*i.e.*, bicubic degradation) and train the model on REDS [8]. REDS4 [17] (*i.e.*, clip 000, 011, 015, 020) is used as the test set. Second, we train the model on Vimeo-90K [18] with two different degradations: bicubic and blur downsampling (Gaussian blur with $\sigma = 1.6$ followed by subsampling). The testing datasets include Vimeo-90K-T [18], Vid4 [7] and UDM10 [19]. On 8 Nvidia A100 GPUs, it takes about 17 days. For video deblurring, we train the model on two different datasets DVD [12] and GoPro [9]. The training time is about 10 days. We test it on their corresponding testing sets. For video denoising, we train the model on the DAVIS [4] and test it on the corresponding testing set and Set8 [14]. The training time is similar to deblurring. We train all models on 8 Nvidia A100 GPUs. It takes about 16.6 days for video SR and 9.7 days for video deblurring and denoising. For training memory cost, it consumes about 39GB and 29GB for video SR and other two tasks, respectively.

**REDS [8]** REDS is a newly-proposed high-quality ($1280 \times 720$) video dataset for video restoration. It has 270 clips for training and validation. Following [17], we use REDS4 (4 selected representative clips, *i.e.*, 000, 011, 015 and 020) for evaluation and the rest 266 clips for training. This dataset is used for training bicubic video SR.

**Vimeo-90K [18]** Vimeo-90K is a widely-used middle-quality ($448 \times 256$) dataset for video restoration. For video SR benchmarking, it uses 64,612 clips for training and 7,824 clips for testing (denoted as Vimeo-90K-T). This dataset is used for training bicubic and blur-downsampling video SR.

**Vid4 [7]** Vid4 is a classical testing dataset for video restoration. It contains 4 video clips (*i.e.*, calendar, city, foliage and walk). Each clip has at least 34 frames ($720 \times 480$).

**UDM10 [19]** UDM10 is a recent proposed testing dataset for video super-resolution. It contains 4 video clips of various scenes, each of which has 32 frames ($1272 \times 720$).

**DVD [12]** DVD is a widely-used high-quality ($1280 \times 720$) dataset for video deblurring. Blurred images are generated from high fps videos. It has 61 videos (5,708 frames in total) for training and 10 videos (1,000 frames in total) for testing.

**GoPro [9]** GoPro is a popular high-quality ($1280 \times 720$) for image and video deblurring. Similar to DVD [12], blurred images are synthesized based on high fps videos. It is consisted of 22 training clips (2,103 frames in total) and 11 testing clips (1,111 frames in total).

**DAVIS [4]** DAVIS-2017 is a popular middle-quality ($854 \times 480$) dataset for video denoising. It consists of 90 videos for training and 30 videos for testing.

**Set8 [15]** Set8 consists of 8 middle quality ($960 \times 540$) videos (*i.e.*, tractor, touchdown, park_joy, sunflower, hypersmooth, motorbike, rafting and snowboard). It is often used as a testing dataset in video denoising. Following [14–16], we only use the first 85 frames of each video.

**Evaluation metrics.** Following [17, 2, 5, 13, 14], we calculate the metrics on RGB channel for REDS4 [17], DVD testing set [12], GoPro testing set [9], DAVIS testing set [4] as well as Set8 [14], and on the Y channel for Vimeo-90K-T [18], Vid4 [7] and UDM10 [19].

## 2 More Visual Comparison

As shown in Fig. 1, Fig. 2 and Fig. 3, we provide more visual results to show the effectiveness of the proposed RVRT on video super-resolution, video deblurring and video denoising. RVRT generates visually pleasing frames with fine details and sharp edges.