# OpenReview forum: "Recurrent Video Restoration Transformer with Guided Deformable Attention"
_NeurIPS.cc/2022/Conference — NeurIPS 2022 Accept_

### Official Review · Reviewer_Js9S · 2022-06-21

**Rating:** 8
**Confidence:** 5
**Soundness:** 4 excellent
**Presentation:** 4 excellent
**Contribution:** 4 excellent

**Summary:**

In this paper, the authors propose a general recurrent video restoration model for various video restoration tasks, including video SR, video deblurring and video denoising. It divides the video sequence into multiple video clips and deal with them by two strategies: globally, it propagates clip features in a recurrent way to reuse model parameters and save memory; locally, it jointly updates different frame features from one clip in parallel. Besides, since it processes the video as clips, it proposes the guided deformable attention for video clip-to-clip alignment. Extensive experiments on various benchmark datasets show the effective and the generalizability of the model.

**Questions:**

See weakness above.

**Limitations:**

The authors have discussed the limitations and potential societal impact.

**Strengths And Weaknesses:**

The proposed RVRT architecture is novel, effective and technically sound. Unlike previous methods that are either recurrent or parallel, RVRT takes the advantages of both directions and alleviates their corresponding problems. As validated in ablation studies and comparisons with existing methods, RVRT makes a good trade-off. It is much smaller, quicker and more memory efficient than parallel methods, and, still, it achieves state-of-the-art performance on benchmark datasets. RVRT also provides a useful way to tackle the information loss and noise application problems that are inherent to recurrent models, as proved in the supplementary. Furthermore, it proposes a guided deformable attention module that is directly applicable for clip-to-clip alignment, in which relevant features from different frames are aggregated dynamically and efficiently. This module is well illustrated and is proved to be effective in experiments.

Overall, this paper makes a good contribution to video restoration and conducts extensive experiments to support its arguments. It provides an alternative video sequence modelling option, rather than focusing on the design of specific modules. It should also be an interesting paper for the wider community. I think this paper should be accepted.


Pros:

1, It proposes a novel architecture that extracts features locally in parallel and accumulates information globally in a recurrent way. It has many benefits due to reduced sequence length, larger hidden state and local parallel processing.

2, It proposes a one-stage guided deformable attention module for dynamic and global feature aggregation in clip-to-clip alignment. It fits the overall architecture well.

3, The proposed method is evaluated on most standard video restoration benchmarks (8 different datasets), and the performance is solid and convincing. In particular, on deblurring and denoising, it remains the surprising performance of transformer models and outperforms most of its competitor by up to 2.27~2.37dB.

4, The paper writing is fairly good.

Cons:

1, What are the training time and memory cost for different tasks?

2, Typo in the caption of Fig. 2.

3, The comparison of temporal modelling ability is interesting and is suggested to be put in the main paper.

---

> ### Author Response · Authors · 2022-08-02
> **Adding more training details, fixing typos and better paper organization.**
>
> Thanks for reviewing our paper. In the following we respond to the questions point by point.
>
> >1, What are the training time and memory cost for different tasks?
>
> We train all models on 8 A100 GPUs. The training time and memory cost per GPU are provided in the following table, which will be added to the paper.
>
> | Task                   | Training Time (day) | Training Memory (MB) |
> |:-----------------------|:-------------------:|:--------------------:|
> | Video Super-Resolution |        16.6         |        39019         |
> | Video Deblurring       |         9.7         |        29263         |
> | Video Denoising        |         9.7         |        29269         |
>
>
> >2, Typo in the caption of Fig. 2.
>
> Thanks for pointing that out. We have fixed the typo.
>
> >3, The comparison of temporal modelling ability is interesting and is suggested to be put in the main paper.
>
> Thanks for your kind advice. We will move it from the supplementary to the main paper.

---

> > ### Comment · Reviewer_Js9S · 2022-08-07
> > **After rebuttal**
> >
> > I appreciate your responses. All of my concerns have been addressed by the response. After seeing the rebuttal and other review comments, I would like to keep my initial score.

---

### Official Review · Reviewer_zj8L · 2022-07-11

**Rating:** 6
**Confidence:** 4
**Soundness:** 3 good
**Presentation:** 2 fair
**Contribution:** 3 good

**Summary:**

This work proposed a recurrent video restoration transformer. It divides the video into multiple clips and the correlations among clips and inside clips are explored jointly. Within each clip, different frame features (2 frames in this work) are updated with implicit feature aggregation. Across clips, guided deformable attention is proposed to perform clip-to-clip alignment. Experimental results demonstrate that the proposed method achieves SOTA results on video SR, delurring, and denoising.

**Questions:**

The authors are suggested to give responses for my concerns about the proposed GDA module. Can it be directly plugged into benchmark video restoration networks (i.e., replacing their original alignment modules) to further improve their performance? If yes, the impact of this work will be great.

**Limitations:**

Yes, the authors have addressed their limitations.

**Strengths And Weaknesses:**

Strengths:
1.	The proposed guided deformable attention for clip-to-clip alignment is the main technical contribution of this work. The ablation study also demonstrates the effectiveness of the proposed module.
2.	The proposed method can efficiently utilize long-term correlations
3.	The proposed method works well on multiple video restoration tasks.

Weaknesses:
More explanations about the proposed module are expected. For example, from table 3, the MLP module is essential for high performance (0.17 dB gain). Why? If we introduce the MLP module to DCN, will it outperform the proposed GDA?  In other words, the gain of GDA over DCN comes from which part of GDA? In addition, since GDA utilizes flow for guidance, a more fair comparison with GDA should be the flow-guided DCA, which was utilized in  BasicVSR++.

In Eq. (5), the predicted M offsets are averaged. Why? In DCN, we utilize multiple offsets for the same pixel and it is demonstrated to be more effective than the single-offset-based optical flow (each pixel only has one offset). What is the value of m in experiments?  Are the results sensitive to different settings of m?

As shown in Table 7, the denoising performance of the proposed method is worse than the compared methods when the sigma is smaller than 20. Why?

---

> ### Author Response · Authors · 2022-08-02
> **Adding more explanation on GDA, offset number and results.**
>
> Thanks for reviewing our paper. In the following we respond to the questions point by point.
>
> >1, More explanations about the proposed module are expected.
>
> >(1) For example, from table 3, the MLP module *** Why?
>
> As discussed in L210 and L258 of the paper, MLP plays an essential role since the deformable attention part only aggregates information spatially and does not allow inter-channel interaction. To be more specific, in GDA, features from different locations are added together by weighted sum (the predicted weight for each location is just a scalar). Therefore, one channel cannot interact with other channels. Our solution is to add a MLP after deformable attention, so that information from different channels are aggregated and interact with each other.
>
> >(2) If we introduce the MLP *** of GDA?
>
> We believe deformable attention and MLP in GDA are inseparable. Deformable attention plays the main role in alignment, because it can aggregate information from multiple relevant locations. As analyzed above, MLP also plays an important role (although less important compared with deformable attention), because it aggregates information from different channels. Both of these two parts are necessary for good performance.
>
> We further conduct ablation studies in the following table to quantitatively evaluate the effectiveness of these two parts.
> As we can see, when we remove deformable attention or MLP, the performance drops by 4.17dB and 0.27dB, respectively. This indicates that deformable attention plays the main role in GDA.
>
> Besides, according to your advice, we add the MLP to DCN for better comparison between deformable attention and deformable convolution. DCN + MLP only brings a minor PSNR improvement of 0.02dB and is still 0.15dB worse than GDA (deformable attention + MLP), which demonstrates that the gain of GDA mainly comes from deformable attention.
>
> |Method|Deformable Attention/Convolution|MLP|PSNR|
> |:-:|:-:|:-:|:-:|
> |DCN + MLP|√|√|31.95|
> |DCN（original)|√||31.93|
> |GDA (no MLP)|√||31.83|
> |GDA (no attention)||√|28.68|
> |GDA (proposed)|√|√|32.10|
>
> >(3) In addition, since GDA utilizes flow *** in BasicVSR++.
>
> For DCN, we used flow guidance for fair comparison (similar to BasicVSR++). Removing the optical flow guidance leads to a significant PSNR drop of 0.91dB for DCN. We will clarify that in the paper.
>
> >2, In Eq. (5), the predicted M offsets *** different settings of m?
>
> In Eq. (5) of the paper, the average of $M$ (offset number) is only used for updating the optical flows in different layers, which can lead to better performance, as shown in Table 3 of the paper. For deformable attention, as indicated in Eq. (7) and (8), we still sample and utilize multiple offsets.
>
> As described in L234, $M$ is set as 9 in experiments. To investigate the impact of $M$, we conduct an ablation study in the following table. As one can see, larger $M$ may result in better performance at the cost of more testing memory and time, since that GDA can utilize more information from more locations. However, when $M$ is very large (e.g., $M=49$), the performance may become worse, possibly because there might be too many irrelevant locations. Therefore, we choose $M=9$ for a trade-off of performance, memory and speed.
>
> |Offset Number|1|9|25|49|
> |:-|:-:|:-:|:-:|:-:|
> |PSNR (dB)|30.03|32.10|32.21|31.85|
> |Testing Memory (MB)|983|1036|1141|1423|
> |Testing Time (ms)|128|143|194|258|
>
> >3, As shown in Table 7, *** smaller than 20. Why?
>
> When sigma is large, the video is heavily corrupted and often requires more spatio-temporal information (or larger receptive fields) for restoration (see Ref. 1 and 2). Based on the recurrent transformer architecture and guided deformable attention, RVRT is effective in long-range dependency modeling. It can utilize information from more frames (temporally) and larger areas (spatially), leading to good performance for large sigma. When sigma is small, it is relatively easy to restore the video from a limited receptive field size. In this case, VRT performs better, possibly due to a larger model size (18.4M) than RVRT (12.8M). To prove it, we show the performance of RVRT when it has a similar number of parameters to VRT (by increasing channel sizes). As one can see, RVRT outperforms VRT for all sigmas under fair comparison.
>
> |$\sigma$|#Para (M)|10|20|30|40|50|
> |:-|:-:|:-:|:-:|:-:|:-:| :-:|
> |VRT|18.4|40.82|38.15|36.52|35.32|34.36|
> |RVRT|12.8|40.57|38.05|36.57|35.47|34.57 |
> |RVRT (large)|18.3|40.91|38.40|36.82|35.72|34.76|
>
> Ref:
>
> [1] Weighted nuclear norm minimization with application to image denoising, CVPR2014
>
> [2] Beyond a gaussian denoiser: Residual learning of deep cnn for image denoising, TIP2017
>
> >4, Can GDA be directly  … the impact of this work will be great.
>
> Yes. It can be used as a plug-and-play module for alignment in most restoration networks. It can also be used for other tasks such as reference SR and cross-modal learning. The codes will be made publically available.

---

> ### Comment · Reviewer_zj8L · 2022-08-08
> **The authors have addressed all of my concerns.**
>
> Thank the authors for their detailed responses. My concerns have been well addressed. I recommend acceptance of this work.

---

### Official Review · Reviewer_CJyZ · 2022-07-11

**Rating:** 6
**Confidence:** 5
**Soundness:** 3 good
**Presentation:** 3 good
**Contribution:** 3 good

**Summary:**

This paper tries to integrate the advantages of parallel methods and recurrent methods, and proposes a recurrent video restoration transformer (RVRT). RVRT first divides the video into fixed-length video clips and refines the subsequent clip feature based on the previously inferred clip feature and the old features of the current clip from shallower layers. Within each clip, the authors use a self-attention mechanism to fuse the feature. Across the clips, the authors propose guided deformable attention (GDA) to achieve video-to-video alignment. The proposed RVRT achieves state-of-the-art performance in video super-resolution, video deblurring, and video denoising.

**Questions:**

See questions in Weaknesses

**Limitations:**

Yes

**Strengths And Weaknesses:**

- Strengths:

1. Parallel methods and recurrent methods are dominant strategies in the VSR area, and most researchers only adopt one strategy in their work. Therefore, the attempt of adopting these two strategies in a single framework is valuable.

2. From Tab. 5, Tab. 7, Tab. 8, and Tab. 9, RVRT achieves state-of-the-art performance in video super-resolution (BI-REDS/Vid4, BD-Vid4), video deblurring, and video denoising.

- Weaknesses:

1. The effectiveness of the core idea (which takes advantage of parallel and recurrent methods) is not well verified.

    1.1. From Tab. 1, when N=3, the performance is worse than N=2. When N=3, the frames within each clip can aggregate more information (the information of the other 2 frames) than N=2. Besides, it also aggregates more information from the neighboring clips (by aggregating the features of 3 frames). Therefore, it is unreasonable that the performance of RVRT saturates when N=3. More detailed analyses (visualization or experiments) are supposed to be provided.

    1.2. From Tab. 6, when BasicVSR++ is equipped with RSTB, the performance gains 0.26dB and achieves 32.61dB. Compared to RVTB, its performance is only 0.14dB lower with 1.2M fewer parameters. Therefore, it is hard to distinguish the effectiveness of the core idea. What is the performance of BasicVSR++ + RSTB, whose parameters are the same as RVRT?

2. Some results are confusing. Wait for the authors’ explanations/analyses.

    2.1. From Tab. 5, it is clear that the performance of RVRT is better than VRT on Vid4 and REDS. However, on Vimeo-90K-T and UM10 datasets, RVRT achieves inferior performance.

    2.2. From Tab. 7, when sigma is 10 and 20, the performance of RVRT is inferior to VRT. However, when sigma is larger than 30, the performance of RVRT is comparable to or better than VRT.

---

> ### Author Response · Authors · 2022-08-02
> **Adding explanation for model saturation, model comparison and results.**
>
> Thanks for reviewing our paper. In the following we respond to the questions point by point.
>
> >1.1. From Tab. 1, when N=3, the performance is worse than N=2. When N=3, the frames within each clip can aggregate more information than N=2. Besides, it also aggregates more information from the neighboring clips. Therefore, it is unreasonable that the performance of RVRT saturates when N=3. More detailed analyses are supposed to be provided.
>
> There are two possible reasons for the performance saturation when $N=3$. First, within each clip, different frames are implicitly aligned by the self-attention mechanism, which may not be good at dealing with large misalignments. For the REDS4 testing set, the average optical flow magnitude for neighboring frames (e.g., between frame 1 and 2) and odd/even frames (e.g., between frames 1 and 3) are 3.89 and 6.91, respectively. However, the spatial window size of RVRT is set as $8\times 8$ due to memory limitation. Therefore, when $N=3$, RVRT may not be able to deal with large misalignments between the first and the third frame in the clip. One solution to this is to increase the spatial window size or add an alignment module for the local transformer part. We leave it as a future work.
>
> The second reason is that the long-distance optical flows might be inaccurate, as pointed out in L248 of the paper. For $N=3$, we need to compute the optical flow between frame 1 and 6. For the sake of computation efficiency, we derive distant flows based on neighboring flows (e.g., we derive flow between frames 1 and 6 based on intermediate flows $1\rightarrow 2$, $2\rightarrow 3$, $3\rightarrow 4$, $4\rightarrow 5$ and $5\rightarrow 6$), which leads to inaccurate optical flows and hinders the performance improvement. To further validate it, we directly estimate all optical flows. In this case, the PSNR rises to 32.21dB for $N=3$, which shows better performance than $N=2$, at the expense of longer testing time (rises from 143ms to 201ms).
>
> >1.2. From Tab. 6, when BasicVSR++ is equipped with RSTB, the performance gains 0.26dB and achieves 32.61dB. Compared to RVTB, its performance is only 0.14dB lower with 1.2M fewer parameters. Therefore, it is hard to distinguish the effectiveness of the core idea. What is the performance of BasicVSR++ + RSTB, whose parameters are the same as RVRT?
>
> We increase the channel size of “BasicVSR++ + RSTB” from 144 to 156, so that "BasicVSR++ + RSTB" has similar parameter numbers to RVRT. As we can see in the following table, directly increasing the channel size brings little improvement. RVRT is still 0.11dB better than "BasicVSR++ + RSTB", which proves that the performance gain mainly comes from architectural design rather than model size.
>
> | Method|#Param (M)|PSNR (dB)|
> |:-|:-:|:-:|
> | BasicVSR++ + RSTB (channel=144)|9.3|32.61|
> | BasicVSR++ + RSTB (channel=156)|10.7|32.64|
> | RVRT|10.8|32.75|
>
> >2.1. From Tab. 5, it is clear that the performance of RVRT is better than VRT on Vid4 and REDS. However, on Vimeo-90K-T and UM10 datasets, RVRT achieves inferior performance.
>
> We argue that it is possibily due to dataset differences in content and motion distributions. First, different datasets may behave differently in testing due to different video contents. Second, as shown in the following table, compared with VRT, RVRT performs differently on fast/medium/slow motion videos in Vimeo-90K. This indicates that the final performance may also vary with motion conditions.
>
> |Method|fast|medium|slow|
> |:-|:-:|:-:|:-:|
> |VRT|41.44|38.42|34.98|
> |RVRT|41.25|38.37|35.07|
>
> >2.2. From Tab. 7, when sigma is 10 and 20, the performance of RVRT is inferior to VRT. However, when sigma is larger than 30, the performance of RVRT is comparable to or better than VRT.
>
> When sigma is large, the video is heavily corrupted and often requires more spatio-temporal information (or larger receptive fields) for restoration (see Ref. 1 and 2). Based on the recurrent transformer architecture and guided deformable attention, RVRT is effective in long-range dependency modeling. It can utilize information from more frames (temporally) and larger areas (spatially), leading to good performance for large sigma. When sigma is small, it is relatively easy to restore the video from a limited receptive field size. In this case, VRT performs better, possibly due to a larger model size (18.4M) than RVRT (12.8M). To prove it, we show the performance of RVRT when it has a similar number of parameters to VRT (by increasing channel sizes). As one can see, RVRT outperforms VRT for all sigmas under fair comparison.
>
> |$\sigma$|#Para (M)|10|20|30|40|50|
> |:-|:-:|:-:|:-:|:-:|:-:| :-:|
> |VRT|18.4|40.82|38.15|36.52|35.32|34.36|
> |RVRT|12.8|40.57|38.05|36.57|35.47|34.57 |
> |RVRT (large)|18.3|40.91|38.40|36.82|35.72|34.76|
>
> Ref:
>
> [1] Weighted nuclear norm minimization with application to image denoising, CVPR2014
>
> [2] Beyond a gaussian denoiser: Residual learning of deep cnn for image denoising, TIP2017

---

> ### Comment · Reviewer_CJyZ · 2022-08-08
> **After Rebuttal**
>
> Thanks for your responses. The authors address my concerns.

---

### Meta-Review · Area_Chair_bChK · 2022-08-25

**Recommendation:** Accept
**Confidence:** Certain

**Metareview:**

The paper introduces a recurrent video restoration transformer with guided attention, which combines recurrent and parallel methods in some extent. All reviewers found that the proposed method is sound and that experiments are adequate to demonstrate the effectiveness of the proposed method.

**Award:**

No

---

### Decision · Program_Chairs · 2022-09-14

Accept